# Acceleration of U.S. Southeast and Gulf coast sea-level rise amplified by internal climate variability

Sönke Dangendorf [1] ✉, Noah Hendricks[2], Qiang Sun[1], John Klinck [2], Tal Ezer [2], Thomas Frederikse [3], Francisco M. Calafat [4], Thomas Wahl [5] & Torbjörn E. Törnqvist [6]

While there is evidence for an acceleration in global mean sea level (MSL) since the 1960s, its detection at local levels has been hampered by the considerable influence of natural variability on the rate of MSL change. Here we report a MSL acceleration in tide gauge records along the U.S. Southeast and Gulf coasts that has led to rates (>10 mm yr$^{-1}$ since 2010) that are unprecedented in at least 120 years. We show that this acceleration is primarily induced by an ocean dynamic signal exceeding the externally forced response from historical climate model simulations. However, when the simulated forced response is removed from observations, the residuals are neither historically unprecedented nor inconsistent with internal variability in simulations. A large fraction of the residuals is consistent with wind driven Rossby waves in the tropical North Atlantic. This indicates that this ongoing acceleration represents the compounding effects of external forcing and internal climate variability.

Sea-level rise is one of the most severe consequences of a warming climate, threatening hundreds of millions of people living in low-lying coastal communities[1,2]. Globally, MSL has increased by ~1.5 mm yr$^{-1}$ since 1900[3–7], a rise that is unprecedented over at least the last 3000 years[8] and primarily induced by the ocean's thermal expansion and barystatic mass changes due to melting ice sheets and glaciers[7]. Global MSL rise has also been accelerating since the 1960s[6], reaching values of >3 mm yr$^{-1}$ over the satellite era since 1992[9,10].

Along the North American East and Gulf coasts (see Supplementary Fig. 1 for an overview of the study area), a combination of natural[11–15] and man-made[12,15,16] vertical land motion (VLM), sterodynamic sea level (SDSL; this is the combination of global mean steric expansion and changes in ocean circulation)[17–24], and changes in Gravitation, Rotation, and Deformation (GRD) accompanying global barystatic mass changes[7,25] has resulted in MSL trends ranging from 1.7 to 8.4 mm yr$^{-1}$ between 1900 and 2021 (Supplementary Fig. 2). Thus, MSL in this region

has generally been rising faster than the global mean[18,26,27] causing considerable impacts that include exponential increases in nuisance flooding[28,29], increased damages due to major storms such as hurricanes Katrina[30] and Sandy[31], and the prospect of accelerating land loss in the most vulnerable settings[32,33]. These examples illustrate that any further increases in the rate of MSL rise, particularly rapid ones, threaten the national security of the U.S. and hamper timely adaptation measures. Continuous monitoring of the rates of MSL rise is therefore of utmost importance and robust and early detection of accelerations may ultimately help to constrain the uncertainties in projections.

In their most recent interagency sea-level rise report, the National Oceanographic and Atmospheric Administration (NOAA) and the National Aeronautics and Space Administration (NASA)[29] compare their process-based (i.e., based on climate model simulations) near-term projections of MSL rise over the 2020–2050 period with quadratic extrapolations (so-called trajectories) of trends estimated from

[1]Department of River-Coastal Science and Engineering, Tulane University, 6823 St. Charles Avenue, New Orleans, LA 70118, USA. [2]Center for Coastal Physical Oceanography, Department of Ocean and Earth Sciences, Old Dominion University, 4111 Monarch Way, Norfolk, VA 23508, USA. [3]Jet Propulsion Laboratory, California Institute of Technology, Pasadena, CA, USA. [4]National Oceanography Centre, 6 Brownlow Street, Liverpool L3 5DA, UK. [5]Department of Civil, Environmental and Construction Engineering, University of Central Florida, Orlando 32816 FL, USA. [6]Department of Earth and Environmental Sciences, Tulane University, 6823 St. Charles Avenue, New Orleans, LA 70118, USA. ✉e-mail: sdangendorf1@tulane.edu

tide gauge records since 1970 (and satellite altimetry since 1993). The process-based projections consider two types of uncertainties: emission uncertainty, which is captured by five scenarios that range from "Low (0.3 m by 2100)" to "High (2 m by 2100)" global MSL rise, and process uncertainty (i.e., arising from our limited understanding of the different processes). While trajectories for most of the U.S. coastlines agree with NOAA's intermediate scenarios, MSL trajectories along the eastern Gulf coast track (or locally even exceed) the "High" scenario, indicating an acceleration that is at the upper end of both expected emission pathways and ice-sheet sensitivities (which dominate the process uncertainties in projections). However, it remains an open question whether this large acceleration in the observations is a robust feature that points to a high-end trajectory of MSL, unresolved processes in the projections such as (non-linear) VLM, or natural ocean dynamic variability that acts to amplify the climate-driven (hereafter "forced") acceleration in observations[34–38] but not in model projections, whose simulated variability is out of phase with the observed variability. Clarifying these open questions and placing the high observational rates into a historical context of the 20th century is therefore crucial, particularly for short- and mid-term planning and decision making.

## Results

### A recent acceleration south of Cape Hatteras

We assess nonlinear rates of MSL rise along the North American East and Gulf coasts based on 66 tide gauge records from the Permanent Service for Mean Sea Level (PSMSL) covering the period 1900–2021 (Supplementary Fig. 1). Each tide gauge record is gap-filled and corrected for linear VLM based on corrections from the recent literature[7]. We also consider nonlinear VLM for tide gauges along the Louisiana and Texas coastlines that are known to have been affected by fluid withdrawal[16]. We follow and build on an approach first introduced by ref. 16 that infers nonlinear VLM from the local differences to the tectonically relatively stable Florida Panhandle (see "Methods" for further details and validation). We use Singular Spectrum Analysis (SSA)[39] to calculate nonlinear trends representative of frequencies longer than 30 years as shown, for example, by the tide-gauge record at Pensacola (Fig. 1). The rates at Pensacola, after removing VLM, have varied around an average rise of 1.4 mm yr⁻¹ with peaks of about 3.7 mm yr⁻¹ in the 1930s and 2.4 mm yr⁻¹ in the 1970s. Since the early 2000s, however, MSL rates have increased to 11.1 mm yr⁻¹ by the end of 2021. To judge whether this steep increase indeed represents a significant acceleration from its 20th century average rate, a Monte Carlo experiment is conducted. At each location, we first generate 1000 artificial time series with similar noise characteristics as tide-gauge records assuming that MSL variations are temporally correlated even at the lowest frequencies[40,41] (see "Methods"). Then the observed linear trend is added to the noise and SSA-based nonlinear rates are calculated for each artificial time series. Finally, the observed rates at each time step are compared to the rates from the noise experiment representing the bounds of natural variability. If the observed rates are larger than in 95% of the cases from the artificial series, we conclude that a significant acceleration exists.

After correcting MSL for VLM, rates along the North American East and Gulf coasts alternate around average rates of 1 and 2 mm yr⁻¹ depending on location (Fig. 2). There are some features that appear coherently along the entire coast, such as peak rates of 3 to 5 mm yr⁻¹ in the late 1930s and reduced rates in the 1950s. Similarly, all tide gauge records show enhanced rates of MSL rise since the late 1990s. But, in our noise experiment most of these fluctuations do not represent a significant acceleration for locations north of Cape Hatteras (except for Annapolis in the Chesapeake Bay, which we interpret as an outlier, and a couple of stations in Maine and Canada). In addition, at some locations maximum rates appear in the earlier period around 1940, such that the most recent rates are not unprecedented. South of Cape

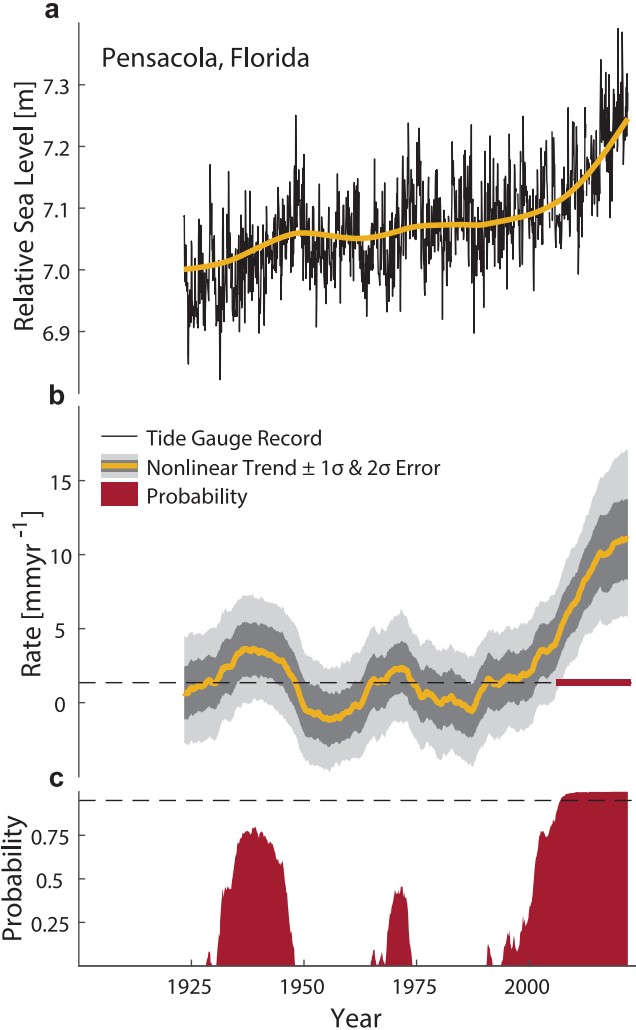

**Fig. 1 | Observed relative mean sea level (MSL) rise acceleration at Pensacola, Florida (PSMSL ID 246). a** Monthly relative MSL record with its nonlinear trend based on Singular Spectrum Analysis (SSA) with a cutoff period of 30 years. **b** Rates of relative MSL rise with the 1- and 2-σ uncertainties from the noise experiment plotted around the observed rates. When the outer bound of the uncertainty envelope exceeds the linear rate (dashed line), the acceleration becomes statistically significant. The period over which this is the case is marked with a dark red line. **c** The corresponding probability function from the noise experiment together with the 95% threshold (dashed line) that determines the statistical significance of an acceleration.

Hatteras, however, all stations exhibit their largest rates at the end of the record between 2018 and 2021. Furthermore, the rates in the south are on average 2–3 times higher than their northern counterparts and they are all significantly different from a long-term correlated random process plus linear trend since the mid-2000s ($P \geq 0.95$) (Fig. 1c, Fig. 2). Such asynchronous MSL behavior across Cape Hatteras has already been noted by other studies at various time scales[19,20,23,42–48] and periods[43], and a similar separation as for the recent acceleration is also visible in the 1970s, where tide gauges experienced accelerated rates south but not north of Cape Hatteras (albeit with a smaller amplitude).

The detection of a significant acceleration extends earlier findings for the altimetry period[38,45] but contrasts with refs. 18,26, who reported an acceleration hotspot between the 1970s and the end of the 2000s farther north in the Mid-Atlantic Bight and the Chesapeake Bay north of Cape Hatteras. There are several potential reasons for this. First, we use a more conservative noise model based on long-term correlations for the evaluation of the statistical significance of the acceleration,

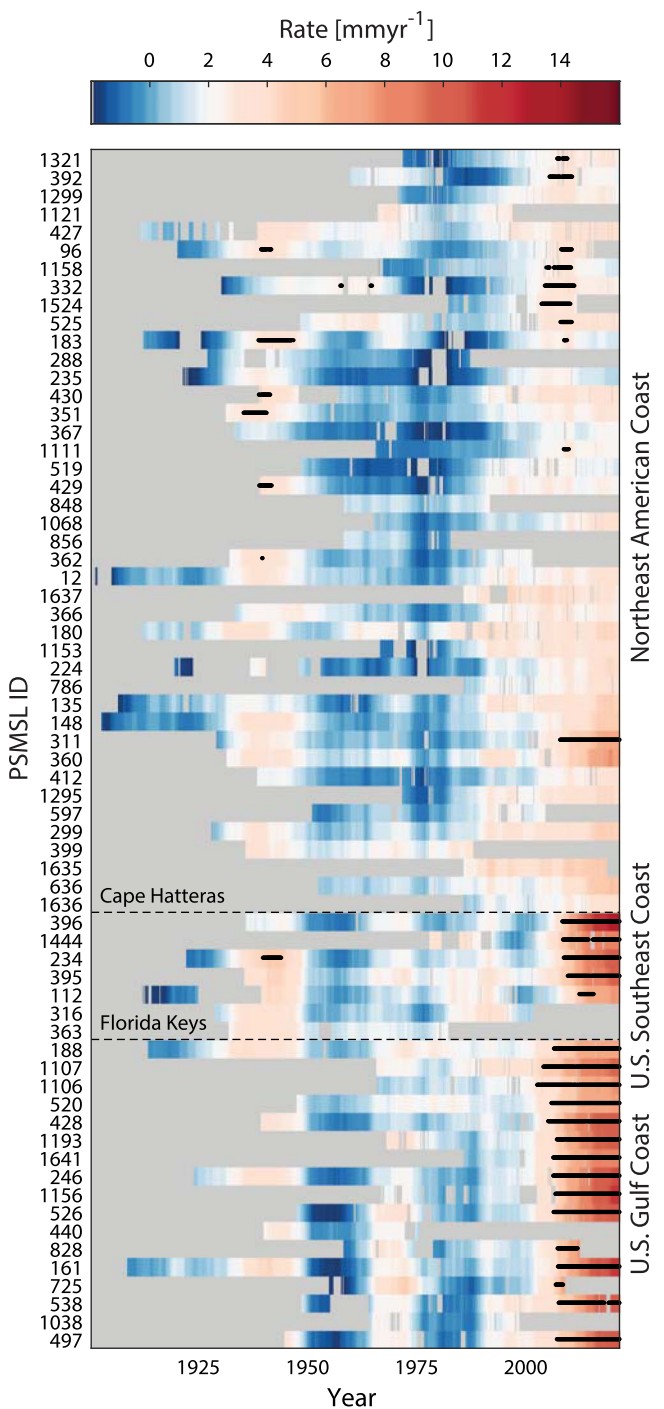

**Fig. 2 | Rates of mean sea level (MSL) rise along the North American East and Gulf coast since 1900.** Color shadings represent the rates of (vertical land motion corrected) MSL with tide gauges arranged from Texas to Newfoundland (bottom to top) following the coastline. Rates that represent a significant acceleration from the mean rate are marked with black dots. The two dashed lines mark Cape Hatteras and the Florida Keys.

while ref. 18 only applied an autoregressive model of the order 1. Second, the approach used in ref. 18 was based on linear trend differences between two consecutive and non-overlapping 20-year periods with the most recent window spanning 1970–1989 and 1990–2009. Thus, they did not consider the high rates in the 1940s (that are also visible in their Fig. 4a), which illustrates that recent rates in this region are not unprecedented during the 20th century. Third, during the 12-year period since ref. 18 (their data ended in 2009, while

data in ref. 26 ended in 2011) no further steepening of the rates has been observed. Instead, recent studies have indicated a southward shift of high MSL rates toward the South Atlantic Bight between 2010 and 2015 from satellite altimetry[38,45,47]. This indicates that internal variability plays an important role in the detection of acceleration "hotspots" along this coastline, a finding that will be further discussed below.

## Physical causes of the acceleration south of Cape Hatteras

There are multiple possible drivers for the recent acceleration south of Cape Hatteras. The global MSL acceleration over the past decades is dominated by increased mass loss from the Greenland and Antarctic ice sheets[9]. However, the associated spatial GRD fingerprints are smoother and do not show an abrupt separation at Cape Hatteras[7] (see also Supplementary Fig. 3). Other potential factors might be local atmospheric pressure and wind changes or contributions through river discharge anomalies[49], but their signatures in coastal MSL are generally an order of magnitude smaller[34,36] and show no sign of a recent acceleration (Supplementary Fig. 3). This therefore leaves only open ocean SDSL changes[23] or unresolved non-linear residual VLM[16] (e.g., due to uncertainties in our VLM corrections) as potential drivers of the recent acceleration.

To get a better idea whether residual VLM or SDSL changes are major drivers, we assess observations from satellite altimetry. As satellite altimetry measures MSL relative to the Earth's center of mass, it is unaffected by any VLM. In addition, it gives a more comprehensive picture of the spatial structure of MSL variability, which may provide further indications about the processes involved. We therefore fit quadratic coefficients to both satellite altimetry and tide gauges over their overlapping period since 1993 (Fig. 3a). Positive coefficients, indicating acceleration, are visible on the continental shelf stretching from the western Gulf of Mexico along the U.S. East Coast up to Cape Hatteras. The acceleration coefficients are qualitatively consistent between satellite altimetry and tide gauges, which implies that non-linear residual VLM can be ruled out as a driver of the region-wide acceleration. We further note that the acceleration also extends off-shore into the Caribbean Sea and the North Atlantic Ocean and becomes most pronounced (>1 mm yr$^{-2}$) in the Subtropical Gyre region. The pattern basically mirrors the major mode of gyre-scale ocean dynamic variability (i.e., related to the expansion/contraction of the ocean) previously identified by ref. 37. This is further underpinned by a similar analysis of steric height[50] of the upper 2000 m in the region (Fig. 3b). Importantly, steric height fields indicate that the pronounced acceleration is primarily induced by an expansion of the Subtropical Gyre, while the central Gulf of Mexico shows a deceleration. This, together with the knowledge that MSL varies coherently along the coast south of Cape Hatteras (Fig. 2), implies that the acceleration has likely been generated offshore[23,36,46]. We will return to this hypothesis below.

## SDSL in observations and historical climate model simulations

The assessment of individual MSL components above has already shown that the disagreement between observational trajectories and process-based NOAA/NASA projections in the Gulf of Mexico cannot be explained by high ice-sheet sensitivities or unresolved VLM. Rather, the acceleration seems to be consistent with an ocean-dynamic signal (Fig. 3). In a next step, we therefore aim to check how well the observations compare to the SDSL signals simulated by climate models used in the projections.

To isolate the coastal SDSL signal, we first remove an estimate of the GRD effects related to contemporary barystatic changes[7] and the inverted barometer effect from each tide-gauge record (Methods). The corresponding residual rates (now primarily indicative of coastal SDSL changes) are highly coherent throughout the study area with a cluster of particularly high correlations (often exceeding 0.9) along

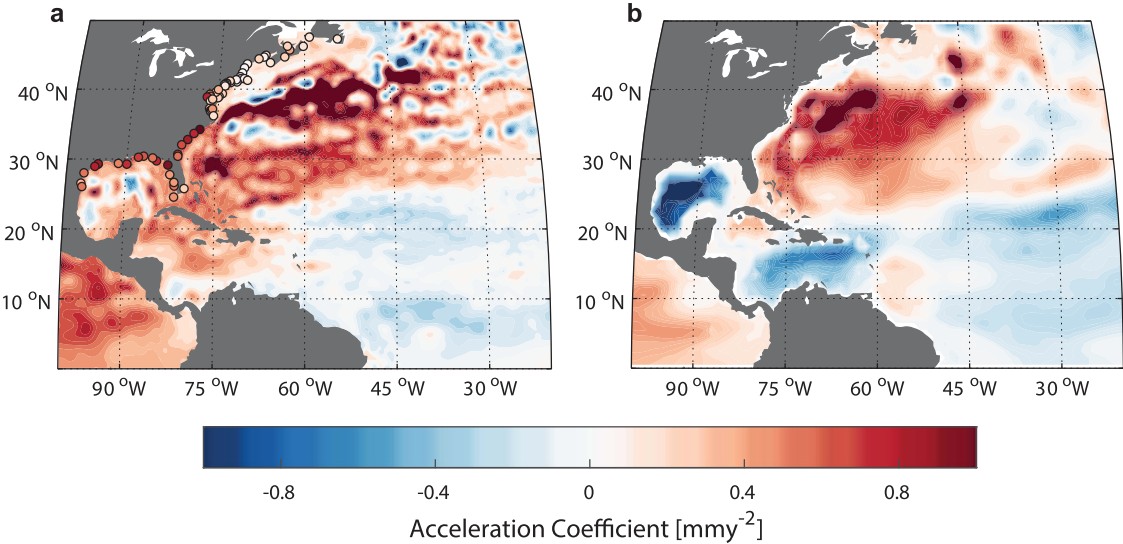

Fig. 3 | **Acceleration of mean sea level (MSL) change rates from tide gauges, satellite altimetry, and steric height over 1993–2020. a** Quadratic coefficients fit to satellite altimetry and tide-gauge records (circles; after removing vertical land motion and inverted barometer contributions; see Methods). **b** Same as (**a**), but for steric height based on gridded fields derived from in-situ temperature and salinity observations from ref. 50.

the U.S. Southeast and Gulf coasts (Supplementary Fig. 4). This and the fact that the acceleration is geographically limited to the south leads us to do the following assessments with a median index over all stations (Methods) south of Cape Hatteras. The index is shown in Fig. 4a. The removal of the GRD and inverse barometer effects reduces the total trend at the lowest frequencies, but barely affects the decadal fluctuations in the rates including the recent acceleration. We compare this residual SDSL signal to an SDSL ensemble from historical simulations of the Coupled Model Intercomparison Project (CMIP) 5[51] and 6[52] (that have also been deployed in the recent NOAA/ NASA projections) and the Community Earth System Model Large Ensemble (CESM LE)[53] that employ estimates of observed (and pro- jected) external radiative forcing (e.g., due to greenhouse gases, solar radiation, volcanic eruptions, land use, and aerosols, Fig. 4, Methods). All three historical model simulation ensembles generally mimic the multidecadal SDSL variability in observations and indicate a recent acceleration. However, the magnitude of the SDSL accel- eration since the 2000s is not replicated by most models, and there is only one CMIP5 ensemble member that simulates an acceleration matching the magnitude of that seen in observations (Fig. 4a, b). This provides evidence that the disagreement between observational trajectories and model simulations in the recent NOAA/NASA report[29] are indeed related to process uncertainties in the SDSL component. We also note, however, that the disagreement is not unique to the most recent period, and a similar mismatch can indeed be seen, for instance, for the peak rates in the 1970s.

It is important to note that while climate model simulations employ observed external forcing, each model starts with its own initial conditions[54]. This means that internal variability varies across ensemble members and usually does not match the phase of observed variability. Given the relatively small number of ensemble members in CMIP5 ($n = 16$) and CMIP6 ($n = 15$) it is therefore not surprising that an acceleration falls outside the range of simulations. This becomes more obvious when removing the ensemble median—an estimate of the externally forced response[54]—from observations and models (Fig. 4b, c). While the most recent rates are still large (exceeded only in seven (CMIP5) and two (CMIP6) percent of all cases in randomized simula- tions; see Methods), they are no longer significantly different to earlier highs, neither in observations nor in simulations. Rather, the obser- vations now show an oscillating pattern with three peaks of compar- able magnitude in the 1940s, 1970s and at the end of the record,

indicating that the residual variability might be related to internal cli- mate variability. We also note that the externally forced response is remarkably coherent between CMIP5, CMIP6, and CESM LE. This coherence also holds geographically, with the result that the spatial correlation clusters in the residual SDSL rates between individual sites become more distinct across Cape Hatteras after the removal of the forced response (Supplementary Fig. 4b).

### Role of internal variability

As noted above, local atmospheric pressure fluctuations, coastal winds, and river discharge cannot satisfactorily explain the recent acceleration in coastal SDSL and therefore also not the mismatch between observations and the simulated forced response (Supple- mentary Fig. 3). Previous studies have indicated that open ocean wind stress curl variations are an important driver of seasonal to decadal MSL variations along the U.S. Southeast Coast likely through the action of Rossby waves[36,44,46], albeit for shorter time scales and periods than investigated here. To test the role of wind-induced Rossby waves in the recent acceleration, we use a 1.5-layer, reduced gravity model solely forced by wind over the open ocean (see Methods). The model domain extends zonally from the west coast of Africa into the Caribbean Sea (~88°W) and solutions are calculated for each latitude between 14°N and 50°N with latitude-dependent phase speeds. We find a strong latitudinal dependence between the outputs from the reduced gravity model and unforced SDSL variations along the coast (Supplementary Fig. 5) with positive correlations at latitudes south of 20°N and north of 38°N and negative correlations at ~32°N. Maximum correlations ($r > 0.7$) are found with Rossby wave signals entering the Caribbean Sea near 18°N. Previous works that indicated remote open ocean signals as a driver of coastal MSL variability south of Cape Hatteras have usually interpreted the alongshore coherence (Supplementary Fig. 4) as being indicative of a signal that is communicated along the coast southward and into the Gulf of Mexico[23,37]. Thus, it is somewhat surprising to find maximum correlations with Rossby wave signals in the Caribbean Sea and not at latitudes close to Cape Hatteras. However, the finding is consistent with ref. 36, who also used a 1.5-layer reduced gravity model coupled to a coastal model. They demonstrated for tide gauge records at Lewes, Delaware, and Fernandina Beach, Florida, that most variance in coastal sea level can be explained when including Rossby waves from tropical regions. They suggested that the increased westward flow due to Rossby waves, after reaching the western wall, would be

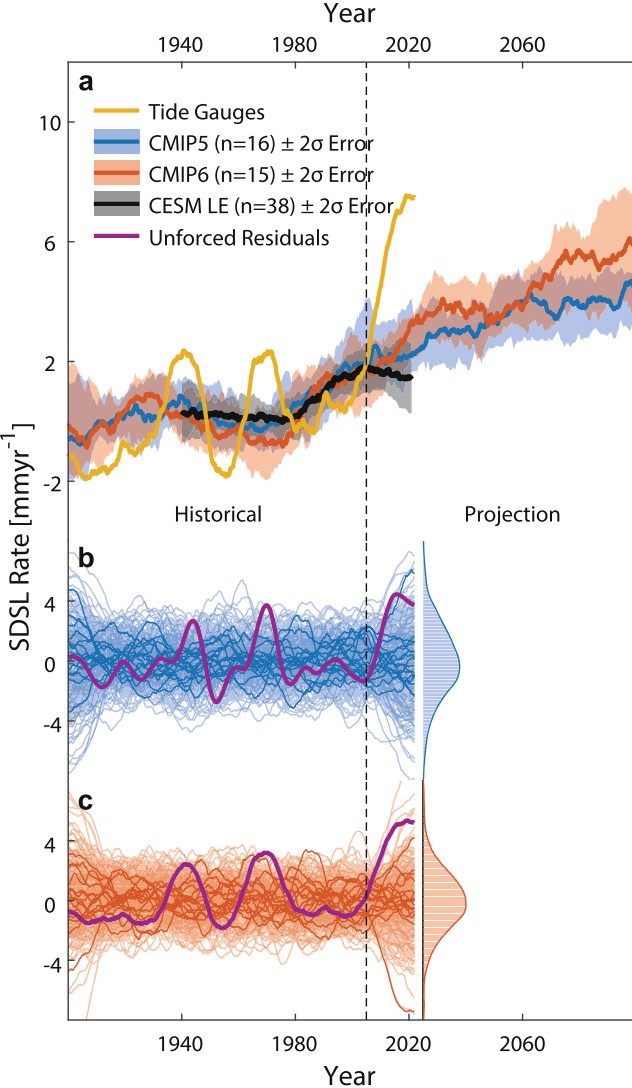

**Fig. 4 | Nonlinear sterodynamic (SDSL) rates along the U.S. Southeast and Gulf coasts from observations and models. a** Nonlinear rates of SDSL averaged from tide-gauge records south of Cape Hatteras after removing contributions from effects of Gravitation, Rotation, and Deformation (GRD) related to barystatic sea-level change, vertical land motion, and the inverted barometer effect. Also shown are the rates simulated by Climate Model Intercomparison Projection (CMIP) models 5 and 6 under historical and future forcing (RCP8.5 and SSP5-8.5, respectively) with the thick lines representing the median and shadings representing 2σ uncertainties calculated over all ensemble members. Also shown are the rates from the 38-member Community Earth System Model (CESM) Large Ensemble. **b** Unforced residual rates after removing the median from all CMIP5 ensemble members (thick lines). The thin lines represent rate variability estimated from 1600 (1500) phase randomized CMIP5 (CMIP6) model simulations (see Methods). The probability density distribution on the right illustrates the range of rates at the end of 2021 from those random samples. **c** Same as in (**b**), but for CMIP6.

absorbed and diverted by the Gulf Stream system[55], subsequently impacting coastal sea level northwards.

If a connection to Rossby waves in the tropics via the Gulf Stream system indeed exists, one would also expect coherence between coastal MSL along the U.S. Southeast and Gulf coasts and MSL in the Caribbean Sea. To test this, we compare the detrended coastal MSL index along the U.S. Southeast and Gulf Coast to the longest of the Caribbean tide gauge records at Magueyes Island, Puerto Rico, over the overlapping period from 1955 to 2021 (Fig. 5a). Both time series show very similar low frequency behavior with peaks in the 1970s and

an acceleration over the past decade. However, the relationship is not fully in phase with maximum correlations (r-0.8 in the median index) appearing once the record at Magueyes Island leads those from the U.S. Southeast and Gulf Coasts by approximately six to eight months (Fig. 5b). As tide gauges only cover the coastal zone, we also calculate correlation maps between a central point in the Caribbean Sea (east of the Caribbean Current) and each other location elsewhere from satellite altimetry at different time lags (Supplementary Fig. 6). At a zero lag, large correlations are confined around the Caribbean Islands with a narrow strip of high positive correlations stretching into the Gulf Stream path. After six to eight months, however, correlations in the Gulf Stream path become larger, extend over larger parts of the Sub-tropical Gyre, and reach into the coastal zones in the Gulf of Mexico. These results support the idea of an advective transfer of density signals from the Caribbean Sea via the larger Gulf Stream system into the Gulf of Mexico and the coastal zones south of Cape Hatteras. This is further supported by recent sensitivity experiments in the adjoint model of the Estimating Circulation and Climate of the Ocean system[56] that pointed to a strong physical linkage between MSL in Charleston, South Carolina, and wind stress forcing (as well as heat and freshwater fluxes) over the Caribbean Sea (maximizing when the latter leads by four to eight months). In line with that, ref. [57] showed that flows through the Gulf of Mexico and the Florida Straits are driven by wind stress curl variations over the tropical North Atlantic. More dedicated ocean model sensitivity experiments will be required to further clarify the impact of Rossby waves onto the Gulf Stream system, the mechanisms by which these signals are transferred into the coastal zones, and what time lags are involved. Those are, however, beyond the scope of this study.

To estimate the integrated gyre-scale effect of wind-forced Rossby waves on coastal sea level south of Cape Hatteras, we isolate the leading modes of variability from the reduced gravity model using principal component analysis (Supplementary Fig. 7, see Methods). The combined signal captures major decadal events such as the highs in the 1940s, 1970s and the large increase over the past decades and its amplitude is dominated by variations originating from the tropics, particularly southeast of the Gulf of Mexico inflow region (Supplementary Fig. 7). Again, this suggests a dominance of advective processes from the south leading to an expansion of the larger Subtropical Gyre region (Fig. 3b). Farther north, where the Gulf Stream is detached from the coast, coastal MSL has rather been linked to density anomalies in the Subpolar Gyre (amongst other factor such as local coastal alongshore wind)[22,23] giving a plausible explanation for why the recent acceleration has been limited to the south. When we combine the resulting Rossby wave signals with other internal forcing factors, notably river discharge[49] and coastal winds[20] (Fig. 6), we find correlations of r = 0.8 with unforced SDSL variability. This suggests that a major fraction of the residual SDSL variability is indeed internally forced and that changes in large-scale wind stress curl over the tropical Atlantic have contributed significantly to the recent acceleration as well as earlier peaks in rates of MSL rise (Fig. 6a). Other factors, such as astronomical cycles, may have also contributed to the recent acceleration as well, but their amplitude is very small compared to the processes discussed here[22].

## Discussion

Our results reveal a significant acceleration (P ≥ 0.95) in Southeastern U.S. coastal MSL that extends from Cape Hatteras into the western Gulf of Mexico. This acceleration has a primarily sterodynamic origin, extends offshore into the Subtropical Gyre and Caribbean Sea, and exceeds historical simulations and projections from climate models. However, we show that this exceedance likely represents a super-position of an externally forced acceleration predicted by climate models (~40%, see Fig. 4a) plus large internal North Atlantic decadal variability that is out of phase with climate model simulations (~60%,

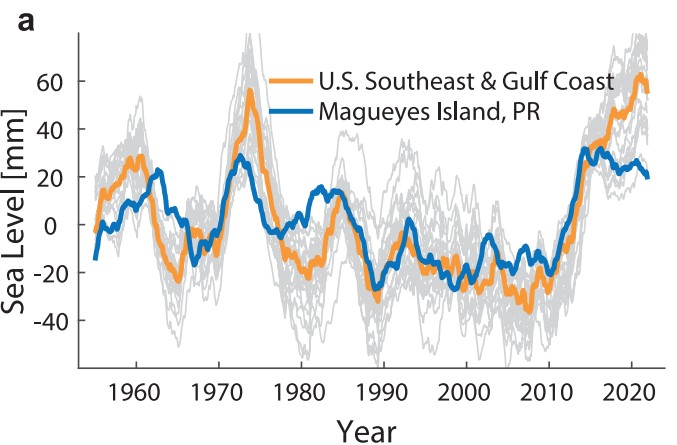
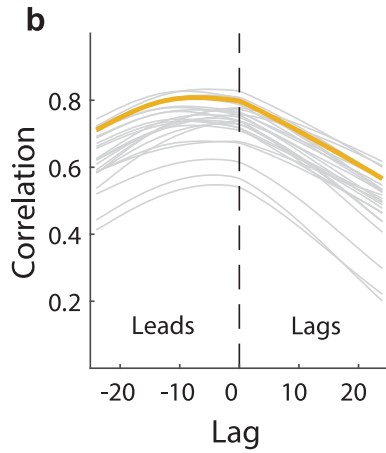

**Fig. 5 | Comparison between mean sea level (MSL) variability along the U.S. Southeast and Gulf coasts and the Caribbean Sea. a** Individual linearly detrended tide gauge records over their overlapping period from 1955 to 2021. All time series have been smoothed with a moving average filter of 48 months. **b** Lead-lag correlation analysis; the left side indicates correlations when the Caribbean Sea leads, while the right side indicates lagged correlations.

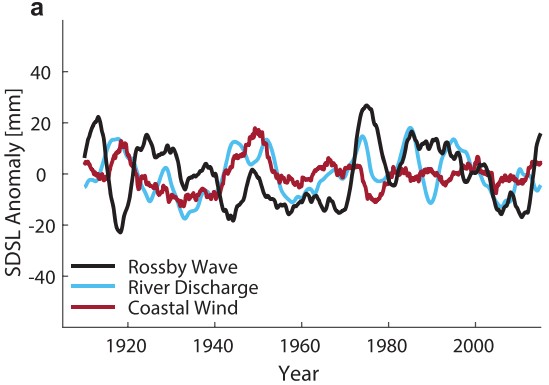
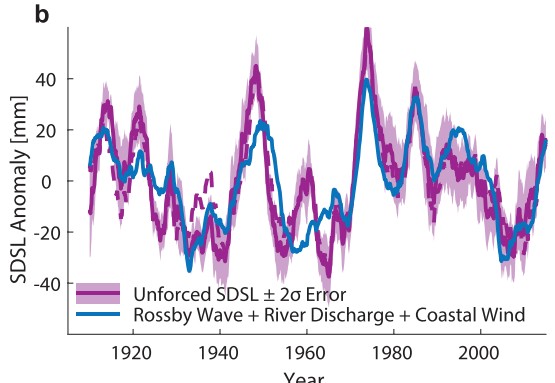

**Fig. 6 | Unforced residual sterodynamic (SDSL) changes versus internal climate variability along the U.S. Southeast and Gulf coasts. a** Individual processes that contribute to coastal SDSL variability averaged (with a median) over all stations south of Cape Hatteras. **b** Residual SDSL index averaged (with a median) over all stations south of Cape Hatteras after removal of forced variability from Climate Model Intercomparison Projection (CMIP) 5 (dashed line) and 6 (solid line) models (i.e., unforced SDSL). The shading indicates the inter-station variability for stations used to produce the index as a 2σ standard error. Overlaid is an estimate of internal climate variability derived from the sum of the three processes in (**a**) (see Methods). All time series are plotted over the overlapping period 1910–2015 and have been smoothed with a moving average filter with a cutoff period of 48 months. The correlation between unforced SDSL and the estimate of internal variability is $r = 0.81$ (CMIP5) and $r = 0.8$ (CMIP6), respectively.

see Fig. 4b, c). We demonstrate that most of the internal variability in coastal MSL is coherent with open ocean wind stress forcing through westward propagating Rossby waves in the tropical North Atlantic that may affect variability in the inflow of water masses into the Caribbean Sea, the Gulf of Mexico, and ultimately the Subtropical Gyre as a whole[55–57]. Showing peak-to-peak variations of ~45 mm (~25 mm due to open ocean wind stress forcing with additional contributions from coastal longshore winds and river discharge) on muti-year timescales such internal variability may either mask or amplify externally forced trends and acceleration along this coastline. It is therefore likely that the MSL rates along the U.S. Southeast and Gulf Coast will return to the average rates projected by climate models within the next decade or so. It also means that there is currently no evidence for a trajectory that follows a high-end projection related to high emission scenarios and high ice-sheet sensitivities. Our results imply that the early detection of acceleration signals, which are needed for near-term planning and decision-making, still represents a major challenge and that comparisons with climate model projections, specifically locally, need to be undertaken with care. More generally, our findings highlight the critical role of a mechanistic understanding of MSL accelerations at the regional scale and its importance for sea-level projections.

## Methods

### Tide-gauge data

We use 66 monthly tide-gauge records along the North American East and Gulf coasts from the online portal of the PSMSL in Liverpool[58], listed in Supplementary Table 1. The selection of tide gauges depends on their availability from the Kalman Smoother[4] and hybrid reconstructions[6] for gap-filling purposes (see below).

### Gap-Filling

The tide gauges have different recording periods and contain gaps (Fig. 2) that hamper analyses such as SSA, Maximum Covariance Analysis, or the calculation of regional indices. We apply an adjusted version of the Data Interpolating Empirical Orthogonal Function (DINEOF) algorithm introduced by ref. 59. Conventional DINEOF can be used to extract EOFs from an incomplete data matrix by using zero as an initial guess for missing values for the calculation of EOF number 1. This EOF is then used as a new guess and the procedure is repeated iteratively until predictions for data gaps are no longer improved. To determine the quality of the gap-filling, 5% of data are retained and the Root Mean Square Error between observations and DINEOF predictions is used as a measure of performance. Different from the classic

DINEOF algorithm, we here start with initial guesses from a local Kalman Smoother[4] and hybrid reconstruction[6]. In our application, 10 iterations were required to reach a Root Mean Square Error of 2.7 cm. We note that, while we use gap-filled records for some of the data analysis approaches, results, such as rates of MSL rise, are usually only presented for data points that stem from original observations.

## VLM correction
Where available, we use the VLM estimates provided by ref. 7, which are based on either Global Navigation Satellite System (GNSS) estimates, or the difference between a tide gauge and nearby satellite altimetry. It is important to note that the ref. 7 estimates already consider corrections for (nonlinear) crustal deformation due to present day mass loss and is consistent with the barystatic GRD ensemble that we use here. These data are available at 55 of the 66 sites. At the remaining locations we use (weighted ensemble mean) estimates from a state-of-the-art Glacial Isostatic Adjustment (GIA) model[60].

It is well known that tide gauges in the Gulf of Mexico are subject to significant nonlinear VLM, likely related to oil, gas, or groundwater withdrawal[16]. These nonlinear changes appear predominantly along the western portions of the U.S. Gulf Coast (Louisiana and Texas). As oceanic decadal MSL variability is known to be coherent among tide gauges in the Gulf, and Pensacola is suggested to be a tectonically relatively stable tide gauge that is affected only by GIA[13], we follow ref. 16 and calculate differences between Louisiana/Texas tide gauges and Pensacola for the estimation of nonlinear VLM. We specifically take barystatic GRD fingerprints[7], GIA[60], and the inverted barometer effect (see below) into account before calculating the difference. Nonlinear VLM is then estimated by fitting a SSA with a cutoff period of 30 years to the differences (Supplementary Fig. 8). As an alternative approach, we also consider the Cedar Key tide gauge farther east as stable reference. The results are shown as a gray dashed line in Supplementary Fig. 8. Nonlinear trends derived from the two different reference gauges provide nearly identical results, underpinning the robustness of the approach. We are aware that potential nonlinearities in VLM may also appear along other U.S. coastlines[12], but the signals are usually an order of magnitude smaller than in the Gulf of Mexico and there is no universally accepted approach to estimate them over the entire 20th century. For instance, ref. 12 indicate an involvement of groundwater withdrawal to potential nonlinearities in the area between southern Virginia and South Carolina. However, the corresponding signals are usually smaller than 2 mm yr$^{-1}$ (Fig. 2 in ref. 12) and, thus, even they have introduced nonlinearities in the (non-observed) past, their impact should be arguably small to the rates discussed here (>10 mm yr$^{-1}$). As an example, the GNSS rates in Norfolk changed from 2.6 mm yr$^{-1}$ (1999–2015) to −1.3 mm yr$^{-1}$ (2010–2015)[12]. Furthermore, any larger non-linearities at individual locations should show up as large deviations in the rates of SDSL rise (e.g., Supplementary Fig. 4), which is not the case. As discussed in the main text, satellite altimetry and tide gauge records agree in terms of the recent acceleration, and any nonlinearities before the altimetry period should not affect the rates since 2010 which are the focus of this study.

Linear rates from our VLM corrections are shown in comparison to observed relative MSL rates for each station in Supplementary Fig. 9. Observed relative MSL trends and VLM show large spatial agreement (r = 0.91) and the VLM corrections reduce the inter-station spread from 1.4 mm yr$^{-1}$ to 0.6 mm yr$^{-1}$. These results underpin the robustness of the estimates used here.

## Barystatic GRD fingerprints
We use the barystatic GRD fingerprints from ref. 7, which are related to barystatic mass changes from glacier melting, ice sheet (Greenland and Antarctica) mass loss, and changes in terrestrial water storage (groundwater depletion, water behind dams, and hydrological

loading). The estimates provide a 100-member ensemble that considers the uncertainties from multiple data and reconstruction sources and spans the period from 1900 to 2020. Here we use the weighted ensemble mean, as each member comes with its individual probability. To capture the entire investigation period from 1900 to 2021 we linearly extrapolate the GRD estimates at each tide gauge based on rates over the most recent 5-year period. Using quadratic extrapolations instead does not significantly alter the results.

## Inverted barometer effect and local wind forcing
The inverted barometer effect is the hydrostatic response of the ocean to sea level pressure fluctuations from the atmosphere[61]. We follow the widely used static approach[61], which assumes that a 1 hPa sea level pressure change induces an ocean response on the order of approximately 1 cm. Coastal MSL is also known to be influenced significantly by local wind forcing resulting in a mostly barotropic response at the coast (Piecuch et al., 2016, 2018). Here, we use estimates from the global barotropic model introduced in refs. 20 and 21. The model is forced with surface winds from the 20th century reanalysis project[62] and only captures the period until the end of 2012. Thus, a multiple linear regression model forced with winds from the National Center for Environmental Prediction's (NCEP) reanalysis[63] is used to extend the model from 2013 to the end of 2021. The regression model follows approaches introduced in refs. 64,65 and estimates the response using a linear multiple stepwise regression model that fits meridional and zonal wind stress from an area of size 4 degrees (lat, lon) surrounding a tide gauge to linearly detrended tide gauge observations. The regression model is then applied to non-detrended wind stress data to capture potential trends resulting from the wind forcing.

## River discharge
River discharge has recently been identified as a significant contributor to coastal SDSL anomalies along the U.S. Southeast and Gulf coasts[49]. We use river discharge time series aggregated for four regions (Gulf of Maine, Mid Atlantic Bight, South Atlantic Bight, Gulf of Mexico) covering the period from 1910 to 2018. To estimate the contribution at each individual tide gauge, we use linear regression of the corresponding regional time series to (detrended) coastal SDSL anomalies. The regression coefficients are in line with those reported in ref. 48, i.e., ranging from 0.1 mm km$^{-3}$ yr to 1.4 mm km$^{-3}$ yr.

## 1.5-layer reduced gravity model
To simulate the propagation of Rossby waves in the North Atlantic, we use a 1.5-layer reduced gravity model following refs. 46 and 66 that is solely forced by wind stress. Changes in sea surface height can be modeled by:

$$\frac{\partial \eta}{\partial t} - C_R \frac{\partial \eta}{\partial x} + \mathbf{R}\eta = -\frac{g'}{g} k \cdot \nabla \times \left( \frac{\tau}{\rho_0 f} \right) \qquad (1)$$

where $\tau$ is the wind-stress vector, $f$ the latitudinally varying Coriolis parameter, $g'$ the reduced gravity, $R$ the decay rate (due to linear drag), $C_R$ the latitudinally varying propagation speed of Rossby waves, and $k$ the vertical unit vector. We follow ref. 46 and choose a decay rate of 1/R = 1.5 years and a reduced gravity $g'$ = 3 cm s$^{-2}$. We vary $C_R$ per latitude using empirical values from ref. 67 ranging from 17 cm/s at 14°N to 2.5 cm/s at 50°N.

We integrate Eq. (1) from a point, $\mathbf{x_e}$, in the eastern North Atlantic at the coasts of Africa and Europe to the coastal zones of the Americas

(Supplementary Fig. 6a):

$$\boldsymbol{\eta}(\mathbf{x_A},\mathbf{t}) = \frac{g'}{gC_R} \int_{x_e}^{x_A} k \cdot \nabla \times \left[ \boldsymbol{\tau}\left(\mathbf{x'},\mathbf{y},\mathbf{t}+\frac{\mathbf{x_e}-\mathbf{x'}}{C_R}\right)\Big/\rho_0 f \right] \exp\left[\mathbf{R}(\mathbf{x_e}-\mathbf{x'})/C_R\right]$$

(2)

where $\mathbf{x_A}$ is the point at which the solution is wanted.

Solutions are based on wind stress calculated from 10 m zonal and meridional winds from the third version of the 20th century reanalysis[68] covering the period 1900 to 2015. We calculate Rossby waves for each latitude individually from 14°N to 50°N. Supplementary Fig. 10 shows a point-by-point correlation between the 1.5-layer reduced gravity model and sea surface height from satellite altimetry. In agreement with earlier studies[63], we find that wind forced Rossby waves can explain a major fraction of sea surface height variability in the open ocean. As indicated in the main text, we also find a latitudinal structure in the correlations between coastal sea level along the U.S. Southeast coast and outputs from the reduced gravity model (Supplementary Fig. 5a). This indicates an active role of the Gulf Stream in mediating the westward propagating signals into the Gulf of Mexico and the coasts South of Cape Hatteras. To estimate their joint effect on coastal sea level, we isolate the dominant modes from the reduced gravity model outputs west of 50°W using principal component analysis. We use the first three principal components (together explaining ~78% of the Rossby wave signal variability) as predictors in a stepwise regression with the coastal SDSL index south of Cape Hatteras as predictand. Results indicate that most of the signal is explained by forcing from southern regions (Supplementary Fig. 7).

## Statistical analysis

To test the significance of the acceleration we base our noise experiments on the finding that the ocean possesses pronounced inertia leading to response times of hundreds to thousands of years[69]. As a result, MSL time series along the world's coastlines typically exhibit long-term correlations[40,41,70] indicated by Hurst coefficients that are significantly larger than 0.5[40]. The Hurst coefficient is determined using Detrended Fluctuation Analysis[40,71], which is particularly robust even in the presence of pronounced nonlinear trends. Here we estimate the Hurst coefficients for all 66 sites. Consistent with earlier findings[40], we estimate Hurst coefficients on the order of 0.8 ± 0.05 (median ± standard deviation over all sites). As the coefficients are mostly stable across locations, we apply a spatially homogeneous Hurst coefficient of 0.8 for the generation of artificial random time series. However, for each site the original standard deviation of the time series is considered. As described in the main text, we use 1000 randomly generated noise time series at each site that are combined with the linear trend from observations to test how large rates can become just by a linear trend plus natural variability induced by a long-memory process. Statistical significance is than estimated by counting the number of exceedances of observations relative to the noise experiment, using a threshold of $P \geq 0.95$.

## Climate model simulations

We use SDSL estimates simulated by the CMIP5 and CMIP6 model ensembles that have previously been used for MSL projections in the Intergovernmental Panel on Climate Change's (IPCC) 5th and 6th Assessment Reports[72,73]. We extract the sea surface height (zos) and global mean thermosteric sea level change (zostoga) variables and combine them to SDSL estimates. All components are corrected for pre-industrial drifts by removing a quadratic fit to pre-industrial control simulations[74]. The climate model simulations are all forced by the same historical external radiative forcings over the 1900–2005 period. After 2005 we use projections following the RCP 8.5 scenario for CMIP5 and SSP 585 for CMIP6. As the model simulations all stem from different modeling groups around the world, parametrizations may vary. This brings in an additional uncertainty when using the ensemble median or average as an estimate of the externally forced response[54]. A more appropriate externally forced response can be derived from a single model large ensemble that uses the same forcings and parametrizations but varying initial conditions to provide a reasonably sized ensemble of different realizations with differently phased internally variability. We use CESM LE with 38 model members to validate that the externally forced response agrees with those from CMIP5 and CMIP6 (Fig. 4a). As we only have the zos variable from CESM-LE, we include the observed global mean steric MSL curve from ref. 50. The multi-decadal patterns among the three ensemble medians agree very well, indicating that each provides a reasonable estimate of the externally forced response. A list of all CMIP5 and CMIP6 models used in this study can be found in Supplementary Table 2.

Unforced variations are assessed by removing the ensemble median from the CMIP5 and CMIP6 models. As the number of simulations ($n = 16$ and $n = 15$ for CMIP5 and CMIP6, respectively) is still too small to accurately describe the bounds of unforced natural variability, we additionally apply a Fourier-based phase scrambling to the residuals. The phase-scrambling approach[75] preserves the spectrum of the original time series but randomizes the phase such that a large ensemble of artificial time series with the same spectral properties as each individual model run can be produced. Here we model 100 additional realizations for each climate model run. These artificial time series provide the basis for the distribution in Fig. 4b.

## Data availability

The tide gauge data used in this study is publicly available from the Permanent Service of Mean Sea Level (https://www.psmsl.org/), while the GRD fingerprints and VLM estimates at individual locations are accessible from the ref. 7 and/or from the cited literature in the methods section. All CMIP5 and CMIP6 models are available under https://esgf-node.llnl.gov/search/cmip5/ and https://esgf-node.llnl.gov/search/cmip6/, respectively.

## Code availability

Codes for the performance of the singular spectrum analysis are publicly available from https://sites.google.com/a/glaciology.net/grinsted/software/ssatrend-m. The sea level data with contributions of each component as well as codes for the evaluation of the sea level rates have been deposited in the ZENODO database under accession code https://doi.org/10.5281/zenodo.7749568[76]. Codes to produce the figures and the Rossby wave model are available from the corresponding author upon request.

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

## Acknowledgements

S.D. acknowledges the NASA grant 80NSSC20K1241, the National Science Foundation grant ICER-2103754 (as part of the Megalopolitan Coastal Transformation Hub), and David and Jane Flowerree for their endowment funds. T.W. acknowledges the NASA grant 80NSSC20K1241 and the National Science Foundation grant ICER-2103754. We are thankful to Christopher Piecuch and Mead Allison for comments on an earlier version of the manuscript. We acknowledge the World Climate Research Programme's Working Group on Coupled Modeling, which is responsible for CMIP, and we thank the climate modeling groups (listed in Supplementary Table 2 of this paper) for producing and making available their model output. For CMIP the U.S. Department of Energy's Program for Climate Model Diagnosis and Intercomparison provides coordinating support and led development of software infrastructure in partnership with the Global Organization for Earth System Science Portals.

## Author contributions

S.D. designed and performed the research together with N.H. and wrote the first draft of the paper. Q.S. coded the Rossby wave model. T.F. contributed sea-level budget and climate model data. J.K., T.E., and F.M.C. assisted in the design and the interpretation of the Rossby wave model results and the influence on sterodynamic sea level. T.E.T. contributed to the analysis and interpretation of vertical land motion. T.W. assisted with the design and the interpretation of the nonlinear trend analysis. All authors shared ideas and contributed to the writing of the manuscript.

## Competing interests

The authors declare no competing interests.
