## [Peer Review File · Nature Communications]

Acceleration of U.S. Southeast and Gulf Coast Sea-Level Rise Amplified by Internal Climate VariabilityReview of “Acceleration of U.S. Southeast and Gulf Coast Sea-Level Rise Amplified by Internal Climate Variability” by Dangendorf et al.

The study investigates the sea level acceleration observed by tide gauge stations along the US southeast and Gulf coasts. This acceleration is unprecedented over the past 120 years and may have socio-economic implications if persisted.

The topic is of great interest and the methodology is sound. I think the study may interest part of the reader of Nature Communication. However, the paper has been written for expert in sea level research and not for the non-expert interested in the content of Nature Communication as the authors employ many specific terminologies and acronyms. This may not help the broader audience of Nature Communication in my opinion.

In addition, I find some results not well supported. Here is an example. The authors state: *“Importantly, steric height fields indicate that the pronounced acceleration is primarily induced by an expansion of the Subtropical Gyre, while the central Gulf of Mexico shows a deceleration. This, together with the knowledge that MSL varies coherently along the coast, implies that the acceleration has likely been ^[17]~~SEP~~ generated offshore and transmitted into the Gulf of Mexico via coastal waveguides.^[17]”* This is just speculation. The authors did not prove anything here. Another example (L167-168): “ We interpret this spatial structure in terms of two different possible mechanisms.” Why don’t you show a specific diagnostic or a specific experiment to demonstrate which mechanism is at play in your findings?

I have the feeling that this study deserves publication but not in Nature Communication. It would better fit in a more conventional journal where the authors would have enough space to develop the methodology and to support their findings. In my opinion, this would help the authors to read the experts in the ‘sea level’ research.

Review of Nature Communications manuscript NCOMMS-22-38522-T
**Acceleration of U.S. Southeast and Gulf Coast Sea-Level Rise
Amplified by Internal Climate Variability,**

by Dangendorf et al.

November 20, 2020

Summary

The paper intends to explain the significant sea-level acceleration signals in the southeast US and the Gulf coast, which exist in the tide gauge observations after removing deterministic components. The paper tries to relate the presence of sea-level acceleration signals in sea surface height observations with an “internal climate event”, by comparison between simulated Rossby waves with sea surface heights both from satellite altimetry and tide gauges. The analysis of the processes is logical and perhaps novel. However, the justification and verification of observed vertical land motion (VLM) seems to be limited. The VLM processes could be complicated, and their signals could contain trends and episodic jumps due in part to anthropogenic or hazardous origins, depending on the coastal region and the geophysical/climatic processes. The paper concisely describes the approach to separate VLM from geocentric sea-level variations (trends, periodicities and perhaps accelerations), more detailed justifications of their methodology and the corresponding convincing results are needed. Since the accuracy of the VLM for this research seems to be unprecedented, so the ability to quantify vertical land motion (linear, episodic, or nonlinear, and at a much finer spatial scale than the sea-level signals) to perhaps successfully articulate the novelty of this work. can decide whether the research is successful or not. There is one perhaps obvious mistake and confusion in **Fig. 3a**, which seems to underscore the credibility of the paper. The authors demonstrate most of the internal variability in coastal sea level can be explained by the Rossby wave in the region of interest. However, the quantification and contribution of the Rossby wave on sea level variabilities (periodicities, trend, and accelerations) in the research region are arguably not evident or clearly shown. Finally, the quantification, and here the explanations of the origins of regional relative sea-level trend/acceleration, could potentially be impact by low-frequency ocean signals (if they exist in the study region, e.g., 60-year periodicity). The authors are urged to address the above issues.

More specific comments are provided bellow, and the manuscript may be publishable if the authors fully address the following comments and after a major revision.

Comments

- While the logic in the study (abstract, Lines 22-33) is reasonable, first to denote observational evidence of sea level accelerations in coastal region of interest, the current climate models cannot explain this phenomenon after removing the simulated and known climate processes, leading to hypothesis that the residual is related with an internal viability. However, it seems that the authors have not yet consider the potential impact of long-period signals in the ocean for reginal sea-level acceleration estimates

e.g., Chambers D.P., M.A. Merrifield, R. S. Nerem, 2012, Is there a 60-year oscillation in global mean sea level? *Geophysical Research Letters*.

H. Bâki Iz and C.K. Shum, 2020; The certitude of a global sea level acceleration during the satellite altimeter era, *J. Geod. Sci.*

In sum, how do the authors justify the regional acceleration signals (versus global ‘uniform’ acceleration signals) are accurately measured, and at a commensurately adequate spatial scale?

- Main body start from line 33, authors introduce the problem found and the meaning behind the research. The paragraph introduces unusual circumstances i.e., acceleration, which cannot be explained by existing studies. From lines 54 to 62, the paper trying to highlight the problem and the meaning of the research, however, it is not easy to follow, and involved an ambiguous term, in lines 56 and 57, “However, it remains an open question whether this large acceleration is a robust feature that points to a high-end trajectory of MSL” what is the meaning of the high-end trajectory of MSL? Are this term means large values in time series from extrapolations? Please clarify.

- Line 63, the paragraph describes the recent acceleration in the south of Cape Hatters and introduce algorithm to decide whether significant accelerations exist or not. Line 105, the physical cause of the sea level acceleration in Cape Hatteras has been defined, the philosophy is as follows: first, excluding possible contributions from ice sheets in sea level measurements for the acceleration, then eliminate the possibility of accelerations introduced by local atmospheric pressure and wind changes, so, the remaining contributors of the phenomena is either from VLM or internal variabilities. To further identify acceleration from which contributor, the research uses quadratic coefficients from both satellite altimetry and tide gauge since 1993 (overlapping period) to prove the vertical land motions in the overlapping period are linear. There is one obvious problem & confusion in **Fig.3 a**. In the description of Fig.3a “Quadratic coefficients fit satellite altimetry and tide-gauge records (circles; after removing VLM and inverted barometer contributions)”. Again, has long period signals (60 years for example) been considered, or would it impact the estimates?

Fig. 3 | Acceleration of MSL change rates from tide gauges, satellite altimetry, and steric height over 1993-2020. a Quadratic coefficients fit to satellite altimetry and tide-gauge records (circles; after removing VLM and inverted barometer contributions). **b** Same as a, but for steric height based on gridded fields derived from in-situ temperature and salinity observations from Chen et al. (2016).

- Please clarify the meaning of “after removing VLM and inverted barometer contributions for tide gauge data”. Tide gauge data is measuring the relative sea level, satellite altimetry measures the absolute (geocentric) sea level. If the meaning of ‘removing’ in the content is correction, theoretically, both measurements from the corrected tide gauge and satellite altimetry are the same. Thus, the figure seems not adequately support the linearity assumption of the VLM in the region of interest.
- The accuracy and reliability of the Vertical land motion correction used in this research need to be expanded and articulate in detail. Assume the research successfully proved linear trends for vertical land motions since 1993, however, is that possible to prove there are no obvious acceleration for vertical land motion right before the satellite altimetry era? Say from 1985 to 1993? If there are obvious accelerations in the vertical land motion on tide gauge locations from 1985 to 1993 is that conclusion still hold?
- Line 130, the studies trying to isolate coastal natural variability signal. **Fig 5** seems to shows almost all the residuals can be explained as Rossby wave in the south of Cape Hatteras, please add some certifications showing other main internal influences, e.g., ENSO, etc. can be ignored, or influences from other internal variabilities may not be obvious.
- The role of internal variability has been demonstrated (line 157 and onward), authors tried to illustrate the reason of latitude depended on the correlation between the Rossby wave and sea level time-series in Extended Data **Fig. 3**, which doesn’t mean good spatial correlation between Rossby wave and sea-level measurements in U.S. southeast coast. More explanations is needed to unambiguously prove the relationship between residual time series and Rossby waves.
- Conclusion. The authors state that ‘most of the internal variability in coastal sea-level can be explained by wind forced Rossby waves’, please quantify the amplitude and contribution of this mode on sea level variability, as this is supposedly the novelty of the manuscript.
- Extended Data **Fig. 5**. Legend of the figure implies that the study is to extract nonlinear VLMs components in the gulf of Mexico by getting residual between tide gauges in the western gulf of Mexico and Pensacola, Florida, in this case, the Pensacola tide gauge needs a linear VLM, and the structure of corrected relative sea level (Gravitation, Rotation, Deformation (GRD) and GIA, inverted barometer effect have been corrected) at this location need to be the same or similar to corrected tide gauge measurements in the Western (Northern) Gulf of Mexico. The primary issue is that authors perhaps needed to clearly show that the structure of the VLM is linear, which may not the case.
- The VLM separation processes in this paper is the key to enable the conclusions of the manuscript. The manuscript indicated that the VLM separation technique used results from [Frederikse et al. 2020], which addressed global (averaged) sea-level study, which may not need high accuracy of VLM estimations compared with these experiments. Please justify this study have adequately accurate VLM motion at a regional scale (spatial scale).
- Furthermore, not all the tide gauges used in the manuscript are from [Frederikse et al, 2020], corrections for some tide gauges used in the GIA model, are the VLMs from the GIA are linear

or nonlinear? Is it that possible some additional errors will be introduced because of VLMs for tide gauges time series from two different methods? It reinforces the request that accuracy of the proposed separated VLM (methodologies and results) need to be justified, especially in describing nonlinear variations in VLMs, because the accuracy of non-linear VLM corrections for this paper is the key to reliable explanations of the sea-level signal.

- The correlations between simulated Rossby wave and sea level measurements in the spatial domain warrants more interpretations. Based on Extended Data Fig 6, the absolute value of correlations between Rossby wave and satellite altimetry in the region of Florida and Gulf of Mexico are small, however, the title says, “acceleration of U.S. southeast and Gulf Coast sea level rise amplified by internal climate Variability”. Please clarify.

Minor Comments:

- Line 22 "While there is evidence for an acceleration in global mean sea-level (MSL) since the 1960s", This is the first-time appearance of the argument in this paper, so reference is needed.
- Line 29, please clarify the meaning of 'unforced variability', is this means non-climate driven variability?
- Line 264–267. The methodologies described are a bit hard to understand, is this description use a wind model to fit the wind contribution using a detrended tide gauge time series? Is that detrend for the tide gauge reasonable?
- Extended Data Fig. 4. Why are there only 4 principal components shown? But the paper in lines 296 - 297 proposed that the first six principal components have been used. Please clarify the rationale of the analysis.

Reviewer 1:

The study investigates the sea level acceleration observed by tide gauge stations along the US southeast and Gulf coasts. This acceleration is unprecedented over the past 120 years and may have socio-economic implications if persisted.

The topic is of great interest and the methodology is sound. I think the study may interest part of the reader of Nature Communication. However, the paper has been written for expert in sea level research and not for the non-expert interested in the content of Nature Communication as the authors employ many specific terminologies and acronyms. This may not help the broader audience of Nature Communication in my opinion.

We thank the reviewer for their critical evaluation. We have significantly revised the manuscript. We took advantage of the more generous format of Nature Communications to specifically explain specific acronyms and terminology. We note that we follow the terminology that has been proposed by Gregory et al. (2019, <https://link.springer.com/article/10.1007/s10712-019-09525-z>) that has not only been adapted by the community but also the most recent IPCC reports. We are confident that the revised manuscript is much more accessible to a wider audience, comparable to other publications in Nature Communications.

In addition, I find some results not well supported. Here is an example. The author's state: *"Importantly, steric height fields indicate that the pronounced acceleration is primarily induced by an expansion of the Subtropical Gyre, while the central Gulf of Mexico shows a deceleration. This, together with the knowledge that MSL varies coherently along the coast, implies that the acceleration has likely been generated offshore and transmitted into the Gulf of Mexico via coastal waveguides."* This is just speculation. The authors did not prove anything here.

First, we want to point out that we have significantly revised the manuscript. Second, the example above cuts off the references that were already listed behind this sentence. Indeed, the coherence of sea level along this part of the coast has been noted by several authors before. In particular, Volkov et al. (2018, ref. 37 in the main paper) indicate gyre-scale heat divergence as a main driver of coherent sea level variability in this region. Also, Calafat et al (2018, ref. 46 in the main paper) suggest incoming Rossby waves as a main mechanism of seasonal sea level variability that travel, after reaching the western boundary, as coastally trapped signals into the Gulf of Mexico. To underpin our statement, we have added Extended Fig. 4 to the revised manuscript (also shown as Figure R3 below), which clearly underpins our statement. We further removed the last section of the sentence, as our results on the Rossby wave forcing suggest an advective process that is involved in the transfer from the Caribbean Sea into the Gulf. When the reviewer states "some results are not well supported", it is unclear which other results they refer to or if the specific example of the Rossby waves was the only result they referred to.

Another example (L167-168): *"We interpret this spatial structure in terms of two different possible mechanisms."* Why don't you show a specific diagnostic or a specific experiment to demonstrate which mechanism is at play in your findings?

We have significantly revised this section. We acknowledge that it lacked clarity in the previous version. A specific ocean model experiment is beyond the scope of this paper. The section on Rossby waves is meant as an additional indication that the residual variability is coherent with remote wind forcing. We acknowledge in the revised manuscript that more dedicated ocean modelling studies are required to clarify the mechanisms with which open ocean sea level signals enter the coastal zone in this region.

I have the feeling that this study deserves publication but not in Nature Communication. It

would better fit in a more conventional journal where the authors would have enough space to develop the methodology and to support their findings. In my opinion, this would help the authors to read the experts in the 'sea level' research.

We thank the reviewer for their suggestion, but we feel this is up to the editor. Again, we have put major efforts into this study and its revision, and we would argue that it is of major interest to a wider audience, not just sea-level scientists, but coastal engineers and ecologists, plus policy makers involved in coastal management of this large region. Finally, we followed the suggestion of the editor to make use of Nature Communication's flexible length restrictions and did in fact do what the reviewer proposes here by expanding the narrative considerably to make it more accessible to a wider audience.

Reviewer 2:

Summary

The paper intends to explain the significant sea-level acceleration signals in the southeast US and the Gulf coast, which exist in the tide gauge observations after removing deterministic components. The paper tries to relate the presence of sea-level acceleration signals in sea surface height observations with an “internal climate event”, by comparison between simulated Rossby waves with sea surface heights both from satellite altimetry and tide gauges. The analysis of the processes is logical and perhaps novel.

We thank the reviewer for their encouraging words.

However, the justification and verification of observed vertical land motion (VLM) seems to be limited. The VLM processes could be complicated, and their signals could contain trends and episodic jumps due in part to anthropogenic or hazardous origins, depending on the coastal region and the geophysical/climatic processes. The paper concisely describes the approach to separate VLM from geocentric sea-level variations (trends, periodicities and perhaps accelerations), more detailed justifications of their methodology and the corresponding convincing results are needed. Since the accuracy of the VLM for this research seems to be unprecedented, so the ability to quantify vertical land motion (linear, episodic, or nonlinear, and at a much finer spatial scale than the sea-level signals) to perhaps successfully articulate the novelty of this work can decide whether the research is successful or not.

There is one perhaps obvious mistake and confusion in **Fig. 3a**, which seems to underscore the credibility of the paper. The authors demonstrate most of the internal variability in coastal sea level can be explained by the Rossby wave in the region of interest. However, the quantification and contribution of the Rossby wave on sea level variabilities (periodicities, trend, and accelerations) in the research region are arguably not evident or clearly shown.

Finally, the quantification, and here the explanations of the origins of regional relative sea-level trend/acceleration, could potentially be impacted by low frequency ocean signals (if they exist in the study region, e.g., 60-year periodicity). The authors are urged to address the above issues.

We are responding to the criticisms in detail further below in the “Comments” section to avoid too much repetition.

More specific comments are provided below, and the manuscript may be publishable if the authors fully address the following comments and after a major revision.

We thank the reviewer for their encouraging words and are confident that we could properly address all comments.

Comments

While the logic in the study (abstract, Lines 22-33) is reasonable, first to denote observational evidence of sea level accelerations in coastal region of interest, the current climate models cannot explain this phenomenon after removing the simulated and known climate processes, leading to hypothesis that the residual is related with an internal variability. However, it seems that the authors have not yet consider the potential impact of long-period signals in the ocean for regional sea-level acceleration estimates.

e.g., Chambers D.P., M.A. Merrifield, R. S. Nerem, 2012, Is there a 60-year oscillation in global mean sea level? *Geophysical Research Letters*.

H. Bâki Iz and C.K. Shum, 2020; The certitude of a global sea level acceleration during the satellite altimeter era, *J. Geod. Sci.*

We thank the reviewer for their comment. The 60-year oscillation in sea level (and other climatic records) is a controversial topic, not least because – if such an oscillation exists – it would statistically be very challenging to infer it from existing records (Nyquist criteria would require at least 2 cycles (120yrs) to be resolved). In any case, even if we would be able to detect a significant 60-year oscillation, we would also need to provide a physical mechanism for its existence. Tidal cycles and barostatic processes are an order of magnitude too small to explain the observed variability (e.g., Woodworth et al. 2010 (<https://www.sciencedirect.com/science/article/abs/pii/S0278434310002207?via%3Dihub>); Frederikse et al., 2020), which leave ocean sterodynamic processes as the only reasonable cause of this variability. Hence, we are looking for a forcing mechanism of this ocean sterodynamic variability, and in case the forcing is non-local, a mechanism that propagates the sea-level anomalies towards our region of interest. For temperature records in the North Atlantic, most recent research indicates that the Atlantic Multidecadal Oscillation (which captures this 60yr periodicity) only appears to be cyclic and can be explained as an artifact from competing components of external forcing (e.g., volcanic forcing and aerosols). Thus, it cannot be attributed to an internal mode of variability (Mann et al., 2021; <https://www.science.org/doi/full/10.1126/science.abc5810>). For sea-level in the North Atlantic the most likely candidate producing such a signal, would be ocean-atmosphere interaction in the form of wind-driven circulation changes. This was already suggested by Chambers et al. (2012) in their original publication. In paragraph 14 they wrote:

“While there is growing evidence of a near 60-year natural climate oscillation and our analysis indicates that some regions have a strong, quasi 60-year variation in sea level, this alone does not mean that there is a detectable GMSL signal. It is likely that a significant fraction of the multi decadal fluctuations in local and regional sea level represent dynamical adjustments to winds, and resulting fluctuations in the strength of the circulation, propagation of Rossby and/or Kelvin waves, or other effects that when averaged globally do not cause a significant amplitude in GMSL. This likely explains some of the phase differences between gauges within the same basin, notably the earlier phase in tide gauges south of Bermuda in the Atlantic Ocean (Key West, Fernandina) and the phase differences in the western North Pacific.”

In paragraph 16 they further noted:

“Our overarching view is that it is difficult to tell if there is a 60-year oscillation in GMSL based on the tide gauge dataset alone without a better understanding of the redistribution signal. Ignoring the issue of land motion at the tide gauges, which in most locations can be treated as a contribution to the long-term relative sea level trend, we take the view that each tide gauge provides a measure of GMSL fluctuations plus regional sea level variations that do not contribute to GMSL. In that context, it seems most likely that the high amplitude 60-year fluctuations in some of the regions, such as the western North Pacific, the South Pacific, the North Atlantic, and the Indian Ocean (Figure 1 and Table 2) represent wind driven changes associated with an internal climate mode. This would be consistent with recent findings that suggest that multi-decadal sea level variation in the western tropical Pacific is due primarily to trade wind forcing [Merrifield and Maltrud, 2011; Becker et al., 2012]. Other regions also exhibit multi decadal sea level variations that appear to be largely wind-driven and not due to GMSL, such as Fremantle [Feng et al., 2004] and the U.S. East coast [Sturges and Douglas, 2011].”

Our study goes beyond a purely statistical fitting of harmonics (again, whose existence is controversial) and rather takes a process-based approach to disentangle individual forcing agent propagation mechanisms (Rossby waves). We demonstrate that the recent acceleration can be explained by the sterodynamic component with both externally forced and internally generated and likely wind-forced components to it. Whether there is a significant contribution by an internally generated 60-year cycle (that would most likely appear due to ocean-atmosphere interactions) would require extended analyses of control simulations, which goes far beyond the scope of this paper.

Last, we would like to note that, in any case, the impact of a 60-year oscillation would be rather minor and not affect any of our conclusions. This is illustrated in Figure R1 below, which shows the nonlinear rates south of Cape Hatteras before and after the removal of a 60-year oscillation. The contribution of the fitted harmonic is smaller than 1 mm/year. We also show the harmonic fits to tide gauge records along the east coast with data coverage larger than 70% (Figure R2). As one can see the fits vary significantly in terms of both amplitude and phase, underpinning the uncertainties surrounding it. We thus believe that its inclusion would only distract from the main message of our paper.

Figure R 1: Rates of sterodynamic sea-level rise south of Cape Hatteras (average over all stations) before and after fitting a 60-year oscillation to the data. Whether or not including this harmonic does not affect any of the conclusions in the manuscript.

Figure R 2: Harmonic fit to all detrended tide gauge records along the East and Gulf coasts (after removing non-sterodynamic contributions) that cover at least 70% of the entire investigation period (1900-2021). Amplitudes vary from 1-23mm, while the phasing can differ by up to 20 years.

In sum, how do the authors justify the regional acceleration signals (versus global 'uniform' acceleration signals) are accurately measured, and at a commensurately adequate spatial scale? It is unclear to us what the reviewer is pointing at here. The study makes use of 66 highly precise tide gauge records (accuracy in cm scale for hourly measurements), and the acceleration appears coherently in all records south of Cape Hatteras. There is also no "globally uniform acceleration". The persistent acceleration since the 1960s has likely been initiated by ocean circulation changes in the Southern Hemisphere potentially leading to increased ocean heat uptake in that region (Dangendorf et al., 2019), and has been continued by barostatic mass changes over the past decades that leave a regional fingerprint (that ranges from a resulting deceleration near the melt-source to an acceleration in the far field; e.g., Coulsen et al. (2022): <https://www.science.org/doi/10.1126/science.abo0926>).

- Main body start from line 33, authors introduce the problem found and the meaning behind the research. The paragraph introduces unusual circumstances i.e., acceleration, which cannot be explained by existing studies. From lines 54 to 62, the paper trying to highlight the problem and the meaning of the research, however, it is not easy to follow, and involved an ambiguous term, in lines 56 and 57, "However, it remains an open question whether this large acceleration is a robust feature that points to a high-end trajectory of MSL" what is the meaning of the highend trajectory of MSL? Are this term means large values in time series from extrapolations?

Please clarify.

We are sorry about the confusion. Observational trajectories (extrapolations based on observations over the past ~50 years) are in the Sea Level Task Force report compared against the uncertain projections to identify whether anything 'unusual' is happening. To make the point clearer, we have extended the whole paragraph as follows:

"In their most recent interagency sea-level rise report, the National Oceanographic and Atmospheric Administration (NOAA) and the National Aeronautics and Space Administration (NASA)²⁹ compare their process-based (i.e., based on climate model simulations) near-term projections of MSL rise over the 2020-2050 period with quadratic extrapolations (so-called trajectories) of trends estimated from tide gauge records since 1970 (and satellite altimetry since 1993). The process-based projections consider two types of uncertainties: emission uncertainty, which is captured by five scenarios that range from 'Low (0.3 m by 2100)' to 'High (2 m by 2100)' global MSL rise, and process uncertainty (i.e., arising from our limited understanding of the different processes). While trajectories for most of the U.S. coastlines agree with NOAA's intermediate scenarios, MSL trajectories along the eastern Gulf coast track (or locally even exceed) the 'High' scenario, indicating an acceleration that is at the upper end of both expected emission pathways and ice-sheet sensitivities (which dominate the process uncertainties in projections). However, it remains an open question whether this large acceleration in the observations is a robust feature that points to a high-end trajectory of MSL, unresolved processes in the projections such as (non-linear) VLM, or natural ocean dynamic variability that acts to amplify the climate-driven (hereafter 'forced') acceleration in observations³⁴⁻³⁸ but not in model projections, whose simulated variability is out of phase with the observed variability. Clarifying these open questions and placing the high observational rates into a historical context of the 20th century is therefore crucial, particularly for short- and mid-term planning and decision making."

- Line 63, the paragraph describes the recent acceleration in the south of Cape Hatters and introduce algorithm to decide whether significant accelerations exist or not. Line 105, the physical cause of the sea level acceleration in Cape Hatteras has been defined, the philosophy is as follows: first, excluding possible contributions from ice sheets in sea level measurements for

the acceleration, then eliminate the possibility of accelerations introduced by local atmospheric pressure and wind changes, so, the remaining contributors of the phenomena is either from VLM or internal variabilities. To further identify acceleration from which contributor, the research uses quadratic coefficients from both satellite altimetry and tide gauge since 1993 (overlapping period) to prove the vertical land motions in the overlapping period are linear. There is one obvious problem & confusion in **Fig.3 a**. In the description of Fig.3a “Quadratic coefficients fit satellite altimetry and tide-gauge records (circles; after removing VLM and inverted barometer contributions)”. Again, has long period signals (60 years for example) been considered, or would it impact the estimates?

Only to a very small fraction. As pointed out in our response to the more general comments above, we have decided to not include any 60-year oscillation. Fitting such harmonics, particularly without knowledge about their origin, is already highly suggestive for a complete 120 tide gauge record (only just fulfilling the Nyquist criterium) and impossible for the 26-year long satellite records shown in Figure 3a of the main paper. As shown in Figures R1 and R2 the impact on the nonlinear rates is small and neglectable for an acceleration that has been taking place over the past 12 years (i.e., since 2010). Lastly, this would not change any of the conclusions regarding the cause of the acceleration, as the 60-year cycle should affect tide gauge and satellite records both in the same way.

- Please clarify the meaning of “after removing VLM and inverted barometer contributions for tide gauge data”. Tide gauge data is measuring the relative sea level, satellite altimetry measures the absolute (geocentric) sea level. If the meaning of ‘removing’ in the content is correction, theoretically, both measurements from the corrected tide gauge and satellite altimetry are the same. Thus, the figure seems not adequately support the linearity assumption of the VLM in the region of interest.

It is unclear to us what the reviewer is referring to here. To compare tide gauge and satellite altimetry records, one needs to correct tide gauges for the mentioned contributions first. The figure demonstrates that the acceleration is visible in both ‘corrected’ tide gauge and satellite records, which underpins that the acceleration is not primarily caused by unaccounted nonlinear VLM.

- The accuracy and reliability of the Vertical land motion correction used in this research need to be expanded and articulate in detail. Assume the research successfully proved linear trends for vertical land motions since 1993, however, is that possible to prove there are no obvious acceleration for vertical land motion right before the satellite altimetry era? Say from 1985 to 1993? If there are obvious accelerations in the vertical land motion on tide gauge locations from 1985 to 1993 is that conclusion still hold?

First, we have accounted for non-linear VLM at locations which are known to be affected by those (i.e., stations west of Pensacola in the Gulf). There is little evidence for any further locations that are significantly affected by nonlinear VLM. If nonlinear VLM would be an issue, then the tide gauge rates of the steredynamic residual should not show any large-scale coherence, which they clearly do. This is illustrated in a new Supplementary Plot that we add below (Figure R3). It shows the spatial correlations between steredynamic residual rates before and after correcting for the forced response from CMIP6 models. South of Cape Hatteras the records show large coherence with generally consistent variability. Finally, we note that nonlinear VLM in the rates before 1993 should not affect the recent acceleration, which has been observed since ~2010.

Figure R 3: Rates of sterodynamic sea-level rise as observed by tide gauges south of Cape Hatteras (top panel of **a** and **b**). Also shown (lower panels) are the correlation matrices between each individual tide gauge pair ordered from northeast to southwest. Records south of Cape Hatteras are marked by the black box in the lower right corner. Results are shown for total sterodynamic sea-level rise (**a**) and sterodynamic sea-level rise after the removal of the forced response from CMIP6 (**b**).

- Line 130, the studies trying to isolate coastal natural variability signal. **Fig 5** seems to show almost all the residuals can be explained as Rossby wave in the south of Cape Hatteras, please add some certifications showing other main internal influences, e.g., ENSO, etc. can be ignored, or influences from other internal variabilities may not be obvious.

Yes, we have checked whether the residual rates can be explained by any climatic indices, but there was no index that can satisfactorily explain the steric sea level variations south of Cape Hatteras ($r_{\text{ENSO}} = -0.1$; $r_{\text{NAO}} = 0.27$). While it would have been methodologically the much simpler solution than coding a Rossby wave model, we would like to note that there is no 'index' that directly influences sea level. Indices, as the name suggests, can be a first guess of potential tele-connective influences on coastal sea level but they don't directly represent a physical forcing process. Indices like ENSO or NAO can have an imprint on sea level via atmosphere-ocean processes, such as wind stress forcing, the inverse barometer effect, or heat and freshwater fluxes. Here we go beyond the simple comparison to ENSO or NAO by introducing a propagation mechanism that considers a direct physical mechanism via Rossby waves. As the Rossby wave model provides an estimate of the integrated response of the ocean to winds (including the resulting time lags) there will be no other index providing a better fit (in addition to the also considered near-coastal winds and river discharge).

- The role of internal variability has been demonstrated (line 157 and onward), authors tried to illustrate the reason of latitude depended on the correlation between the Rossby wave and sea level time-series in Extended Data **Fig. 3**, which doesn't mean good spatial correlation between Rossby wave and sea-level measurements in U.S. southeast coast. More explanations is needed to unambiguously prove the relationship between residual time series and Rossby waves.

We thank the reviewer for this comment and admit that the section on the Rossby wave was lacking some clarity. As pointed out in the main paper, we found that the agreement between U.S. Southeast/Gulf Coast sea level and the Rossby wave model outputs is largest when considering remote signals that stem from the tropical North Atlantic, in particular the signals from the Caribbean Sea (where the local correlation between the Rossby wave model and SSH exceeds $r = 0.9$). This is somewhat different to other applications, where Rossby wave models are used as diagnostics at a given latitude. Here, however, we interpret the incoming Rossby waves at **different latitudes** as a remote open ocean signal forcing changes in coastal sea level via adjustments in the Gulf Stream transport and the Subtropical Gyre as a whole. This interpretation is principally consistent with Hong et al. (2000), who suggested that coastal sea level along the Atlantic Southeast coast (Fernandina Beach to Lewes) can be interpreted in terms of SSH differences (induced by incoming wind-forced Rossby waves) between Cape Hatteras and the tropical Northwest Atlantic ($\sim 18\text{N}$). They suggest that increased flow into the west is absorbed into the transport of the Gulfstream, and that lower latitudes in the tropics play a major role for that (please see Figure R6 below which shows the explained variance in coastal sea level by including Rossby waves from different latitudes into their model). Our results are very consistent with that interpretation, though we do calculations over a more than three times longer period (their model only spanned 1962 to 1987) and with a focus on the recent acceleration since 2010. What remains unanswered, however, is the question how these signals enter the coastal zone in the Gulf of Mexico, how effective geostrophy along the U.S. Southeast coast (where the Gulf Stream flows very close to the shore) really is at decadal timescales, and what role the Gyre plays. These questions are, however, beyond the scope of this paper and should be addressed in more dedicated ocean modelling experiments with more advanced circulation models. The most important message for this study is that the remaining residual sea level, after removing forced historical climate model simulations, is consistent with the combination of wind-driven variability in the open ocean, river discharge, and near-coastal winds (whereby the remote wind-driven signal via Rossby waves clearly dominates). To make

this clearer, we have significantly expanded the section and tried to be more outspoken about remaining uncertainties.

FIG. 12. Percent of variance accounted for in the computed sea level signals at Fernandina and Lewes as a function of the latitude to which the model calculations extend.

Figure R 4: Figure 12 from Hong et al. (2000).

- Conclusion. The authors state that ‘most of the internal variability in coastal sea-level can be explained by wind forced Rossby waves’, please quantify the amplitude and contribution of this mode on sea level variability, as this is supposedly the novelty of the manuscript.

We thank the reviewer for this suggestion. We have revised the section which now reads as follows: “Our results reveal a significant acceleration ($P \geq 0.95$) in Southeastern U.S. coastal MSL that extends from Cape Hatteras into the western Gulf of Mexico. This acceleration has a primarily steric origin, extends offshore into the Subtropical Gyre and Caribbean Sea, and exceeds historical simulations and projections from climate models. However, we show that this exceedance likely represents a superposition of an externally forced acceleration predicted by climate models (~40%, see Fig. 4a) plus large internal North Atlantic decadal variability that is out of phase with climate model simulations (~60%, see Fig. 4b,c). We demonstrate that most of the internal variability in coastal MSL is coherent with open ocean wind stress forcing through westward propagating Rossby waves in the tropical North Atlantic that may affect variability in the inflow of water masses into the Caribbean Sea, the Gulf of Mexico, and ultimately the Subtropical Gyre as a whole⁵⁵⁻⁵⁷. Showing peak-to-peak variations of ~45 mm (~25 mm due to open ocean wind stress forcing with additional contributions from coastal longshore winds and river discharge) on multi-year timescales such internal variability may either mask or amplify externally forced trends and acceleration along this coastline. It is therefore likely that the MSL rates along the U.S. Southeast and Gulf Coast will return to the average rates projected by climate models within the next decade or so. It also means that there is currently no evidence for a trajectory that follows a high-end projection related to high emission scenarios and high ice-sheet sensitivities. Our results imply that the early detection of acceleration signals, which are needed for near-term planning and decision-making, still represents a major challenge and that comparisons with climate model

projections, specifically locally, need to be undertaken with care. More generally, our findings highlight the critical role of a mechanistic understanding of MSL accelerations at the regional scale and its importance for sea-level projections.”

- Extended Data Fig. 5. Legend of the figure implies that the study is to extract nonlinear VLMs components in the gulf of Mexico by getting residual between tide gauges in the western gulf of Mexico and Pensacola, Florida, in this case, the Pensacola tide gauge needs a linear VLM, and the structure of corrected relative sea level (Gravitation, Rotation, Deformation (GRD) and GIA, inverted barometer effect have been corrected) at this location need to be the same or similar to corrected tide gauge measurements in the Western (Northern) Gulf of Mexico. The primary issue is that authors perhaps needed to clearly show that the structure of the VLM is linear, which may not be the case.

We thank the reviewer for their comment. Our approach is based on Kolker et al. (2011) and Zervas et al. (2013, <https://repository.library.noaa.gov/view/noaa/14751>) and assumes that Pensacola, Florida represents a geologically stable location largely unaffected by any nonlinear VLM. This assumption is reasonable with respect to VLM due to potential oil and gas withdrawals, or deltaic processes that have been active along the Louisiana and Texas coastlines. Thus, the only reasonable remaining factor that might induce nonlinear VLM at Pensacola might be groundwater withdrawals. However, those are usually localized and, to our knowledge, have not been reported for the Pensacola tide gauge. As an additional test we have therefore used the Cedar Key tide gauge at the Florida west coast instead of Pensacola. As can be inferred from Figure R4 and Figure R5 (the updated Extended Data Figure 5; now 8), differences in the (nonlinear) VLM estimates are really minor. Nonlinearities agree very well and differences in linear VLM can be attributed to different record length at the two sites. Again, this evidence is further supported by the fact that the nonlinear sea-level rates are coherent from Cape Hatteras into the Gulf of Mexico. We added more text to the revised manuscript, particularly to the Methods section.

Figure R 5: Linear Trends of VLM as inferred from the difference to Pensacola and Cedar Key. Note that both tide gauges cover slightly different periods (Pensacola starts 1923; Cedar Key in 1938).

Figure R 6: Updated Extended Data Figure 5 (now 8) from the original manuscript: **Nonlinear VLM in the Gulf of Mexico**. Shown are the residuals (colored dots) between tide gauge records in the western Gulf of Mexico and Pensacola, Florida. The tide gauge records have been corrected for GRD, GIA, the inverted barometer effect, and coastal winds before calculating the differences to Pensacola. The corresponding VLM changes, inferred from a SSA with an embedding dimension corresponding to a cutoff period of 30 years (see **Methods**), are shown as thick black lines. As validation the same VLM changes inferred from the differences to the record of Cedar Key, Florida, are also shown (grey dashed line). Both agree very well, indicating that the resulting nonlinear VLM estimates are robust

- The VLM separation processes in this paper is the key to enable the conclusions of the manuscript. The manuscript indicated that the VLM separation technique used results from [Frederikse et al. 2020], which addressed global (averaged) sea-level study, which may not need high accuracy of VLM estimations compared with these experiments. Please justify this study have adequately accurate VLM motion at a regional scale (spatial scale).

We thank the reviewer for their comment. First, we would like to point out that the VLM estimates used in this study do not suffer from inaccuracies that might occur when analyzing huge amounts of data and

neglecting local uncertainties. The dataset used here is based on Frederikse et al. (2020), but the data base used in Frederikse et al. (2020) has been built up over many years with several other studies published along the way: e.g., for the Northeast: Frederikse et al. (2017): subsidence <https://agupubs.onlinelibrary.wiley.com/doi/full/10.1002/2017JC012699>; for the entire Atlantic: Frederikse et al. (2018): <https://journals.ametsoc.org/view/journals/clim/31/3/jcli-d-17-0502.1.xml>; Altimetry minus Tide Gauge: Kleinherenbrink et al (2018): <https://os.copernicus.org/articles/14/187/2018/os-14-187-2018-discussion.html>; Impact of GRD on GNSS: Frederikse et al. (2019): <https://se.copernicus.org/articles/10/1971/2019/se-10-1971-2019-discussion.html>; Nonlinear VLM in the Gulf: Kolker et al. (2011): <https://agupubs.onlinelibrary.wiley.com/doi/full/10.1029/2011GL049458>; GIA correction: Caron et al. (2018): <https://agupubs.onlinelibrary.wiley.com/doi/full/10.1002/2017GL076644>. Thus, our estimate builds on the tremendous effort that multiple researchers have undertaken in this specific region. Second, we feel that the VLM correction has a minor impact on the results of this study as (with exception of the few stations west of Pensacola) VLM is considered to be linear and the acceleration we are looking at happened after ~2010 (such that potential nonlinearities (which seem to be minor based on Figure R3) before that time wouldn't have a direct impact). Nevertheless, to showcase that the VLM correction provides reasonable estimates, we added a new figure (Figure R4 below) to the Supplementary Material. The figure shows that VLM accounts for a majority of the spatial structure in linear Trends in this area. Indeed, the observed linear rates and the VLM correction are spatially highly correlated ($r = 0.91$) underpinning the accuracy of the latter. The inter-station spread of linear trends is halved when removing the VLM correction.

Figure R 7: Observed linear trends in relative sea level (black), VLM (brown), and corrected residual sea level (blue) along the U.S. East and Gulf coasts over the 1900 to 2021 period.

- Furthermore, not all the tide gauges used in the manuscript are from [Frederikse et al, 2020], corrections for some tide gauges used in the GIA model, are the VLMs from the GIA are linear or nonlinear? Is it that possible some additional errors will be introduced because of VLMs for tide gauges time series from two different methods? It reinforces the request that accuracy of the proposed separated VLM (methodologies and results) need to be justified, especially in describing nonlinear variations in VLMs, because the accuracy of non-linear VLM corrections for this paper is the key to reliable explanations of the sea-level signal.

As stated above we have revised the manuscript and added more information on the VLM correction. It is standard in the sea-level community to assume GIA around the Laurentide icesheet being linear over

periods up to a couple of 100 years, so yes, as in any other 20th century sea level study we assume GIA to be linear. Uncertainties in linear VLM should not affect the nonlinear rates in our sea level records. For instance, the uncertainties are likely to explain some of the vertical spread in the rates between individual records (Figure R3, top), but would not have any influence of the temporal pattern. This could only be introduced by nonlinearities, which we specifically take into account for locations in the northwestern Gulf of Mexico following (and expanding on) the approach by Kolker et al. (2011). Along the Atlantic coast, there is very limited evidence for nonlinear VLM. For instance Karegar et al, (2016, <https://agupubs.onlinelibrary.wiley.com/doi/10.1002/2016GL068015>) note that there is no significant difference between geological processes such as GIA and present day vertical rates from GNSS. An exception is the area between Virginia and South Carolina, where groundwater withdrawal has likely contributed to subsidence. However, the signals are usually smaller than 2 mm/yr (see Figure R5 below) and thus, even if they have introduced nonlinearities in the (non-observed) past, their impact should be arguably small with respect to the rates (>10 mm/yr) discussed in this paper. For instance, in Norfolk, VA (which is just outside, but close to the border of our main focus region of the acceleration), they mention that GNSS trends changed from -2.6 (1999-2015) to -1.3 mm/yr (2010-2015). We have expanded on the discussion of these nonlinearities in the revised manuscript.

Figure R 8: (a) Comparison of vertical motion as a function of latitude from GPS and geologic data. Color bar shows the length of time series for individual stations. (b) Spatially averaged GPS, geologic data, and GIA model ICE6G-VM5a (c). Taken from Karegar et al. (2016).

- The correlations between simulated Rossby wave and sea level measurements in the spatial domain warrants more interpretations. Based on Extended Data Fig 6, the absolute value of correlations between Rossby wave and satellite altimetry in the region of Florida and Gulf of Mexico are small, however, the title says, “acceleration of U.S. southeast and Gulf Coast sea

level rise amplified by internal climate Variability”. Please clarify.

We have significantly revised and extended the section on Rossby waves. As noted in the response to the earlier comment on Rossby waves, we feel that the section lacked clarity leading to a misunderstanding in the interpretation of Extended Data Fig. 6 (now 10). This figure only shows the local correlation between SSH and the local output of the Rossby wave model at that specific location. However, as we discuss in the revised manuscript, we interpret coastal sea level in that region in terms of a remote open ocean signal that is propagated by Rossby waves westward and once reaching the western boundary affects the flow of the larger Gulf Stream system. Thus, a local correlation between the Rossby wave model and SSH should not necessarily be expected. This is supported by several other findings in recent literature, which we now added to the section. However, we note that further details on the impact of Rossby waves onto the Gulf Stream system and coastal sea level require more dedicated sensitivity experiments in general circulation models, which is beyond the scope of this study. The revised section now reads as follows:

*“As noted above, local atmospheric pressure fluctuations, coastal winds, and river discharge cannot satisfactorily explain the recent acceleration in coastal SDSL and therefore also not the mismatch between observations and the simulated forced response (**Extended Data Fig. 3**). Previous studies have indicated that open ocean wind stress curl variations are an important driver of seasonal to decadal MSL variations along the U.S. Southeast Coast likely through the action of Rossby waves^{36,44,46}, albeit for shorter time scales and periods than investigated here. To test the role of wind-induced Rossby waves in the recent acceleration, we use a 1.5-layer, reduced gravity model solely forced by wind over the open ocean (see **Methods**). The model domain extends zonally from the west coast of Africa into the Caribbean Sea (~88°W) and solutions are calculated for each latitude between 14°N and 50°N with latitude-dependent phase speeds. We find a strong latitudinal dependence between the outputs from the reduced gravity model and unforced SDSL variations along the coast (**Extended Data Fig. 5**) with positive correlations at latitudes south of 20°N and north of 38°N and negative correlations at ~ 32°N. Maximum correlations ($r > 0.7$) are found with Rossby wave signals entering the Caribbean Sea near 18°N. Previous works that indicated remote open ocean signals as a driver of coastal MSL variability south of Cape Hatteras have usually interpreted the alongshore coherence (**Extended Data Fig. 4**) as being indicative of a signal that is communicated along the coast southward and into the Gulf of Mexico^{23,37}. Thus, it is somewhat surprising to find maximum correlations with Rossby wave signals in the Caribbean Sea and not at latitudes close to Cape Hatteras. However, the finding is consistent with ref. 36, who also used a 1.5-layer reduced gravity model coupled to a coastal model. They demonstrated for tide gauge records at Lewes, Delaware, and Fernandina Beach, Florida, that most variance in coastal sea level can be explained when including Rossby waves from tropical regions. They suggested that the increased westward flow due to Rossby waves, after reaching the western wall, would be absorbed and diverted by the Gulf Stream system⁵⁵, subsequently impacting coastal sea level northwards.*

*If a connection to Rossby waves in the tropics via the Gulf Stream system indeed exists, one would also expect coherence between coastal MSL along the U.S. Southeast and Gulf coasts and MSL in the Caribbean Sea. To test this, we compare the detrended coastal MSL index along the U.S. Southeast and Gulf Coast to the longest of the Caribbean tide gauge records at Magueyes Island, Puerto Rico, over the overlapping period from 1955 to 2021 (**Fig. 5a**). Both time series show very similar low frequency behavior with peaks in the 1970s and an acceleration over the past decade. However, the relationship is not fully in phase with maximum correlations ($r \sim 0.8$ in the median index) appearing once the record at Magueyes Island leads those from the U.S. Southeast and Gulf Coasts by approximately six to eight months (**Fig. 5b**). As tide gauges only cover the coastal zone, we also calculate correlation maps between a central point in the Caribbean Sea (east of the Caribbean Current) and each other location elsewhere from satellite altimetry at different time lags (**Extended Data Fig. 6**). At a zero lag, large correlations are*

confined around the Caribbean Islands with a narrow strip of high positive correlations stretching into the Gulf Stream path. After six to eight months, however, correlations in the Gulf Stream path become larger, extend over larger parts of the Subtropical Gyre, and reach into the coastal zones in the Gulf of Mexico. These results support the idea of an advective transfer of density signals from the Caribbean Sea via the larger Gulf Stream system into the Gulf of Mexico and the coastal zones south of Cape Hatteras. This is further supported by recent sensitivity experiments in the adjoint model of the Estimating Circulation and Climate of the Ocean (ECCO) system⁵⁶ that pointed to a strong physical linkage between MSL in Charleston, South Carolina, and wind stress forcing (as well as heat and freshwater fluxes) over the Caribbean Sea (maximizing when the latter leads by four to eight months). In line with that, ref. 57 showed that flows through the Gulf of Mexico and the Florida Straits are driven by wind stress curl variations over the tropical North Atlantic. More dedicated ocean model sensitivity experiments will be required to further clarify the impact of Rossby waves onto the Gulf Stream system, the mechanisms by which these signals are transferred into the coastal zones, and what time lags are involved. Those are, however, beyond the scope of this study.

To estimate the integrated gyre-scale effect of wind-forced Rossby waves on coastal sea level south of Cape Hatteras, we isolate the leading modes of variability from the reduced gravity model using principal component analysis (**Extended Data Fig. 7**, see **Methods**). The combined signal captures major decadal events such as the highs in the 1940s, 1970s and the large increase over the past decades and its amplitude is dominated by variations originating from the tropics, particularly southeast of the Gulf of Mexico inflow region (**Extended Data Fig. 7**). Again, this suggests a dominance of advective processes from the south leading to an expansion of the larger Subtropical Gyre region (**Fig. 3b**). Farther north, where the Gulf Stream is detached from the coast, coastal MSL has rather been linked to density anomalies in the Subpolar Gyre (amongst other factor such as local coastal alongshore wind)^{22,23} giving a plausible explanation for why the recent acceleration has been limited to the south. When we combine the resulting Rossby wave signals with other internal forcing factors, notably river discharge⁴⁹ and coastal winds²⁰ (**Fig. 6**), we find correlations of $r = 0.8$ with unforced SDSL variability. This suggests that a major fraction of the residual SDSL variability is indeed internally forced and that changes in large-scale wind stress curl over the tropical Atlantic have contributed significantly to the recent acceleration as well as earlier peaks in rates of MSL rise (**Fig. 6a**).”

Minor Comments:

- Line 22 "While there is evidence for an acceleration in global mean sea-level (MSL) since the 1960s", This is the first-time appearance of the argument in this paper, so reference is needed. We are sorry, but the format of Nature journals does no longer allow for citations in the abstract. The citation is given shortly after as part of the first paragraph. It is reference No. 6 (Dangendorf et al., 2019).

- Line 29, please clarify the meaning of 'unforced variability', is this means non-climate driven variability?

Internal and unforced variability mean the same thing. We have adjusted the text to avoid any confusion.

- Line 264–267. The methodologies described are a bit hard to understand, is this description use a wind model to fit the wind contribution using a detrended tide gauge time series? Is that detrend for the tide gauge reasonable?

We thank the reviewer for their comment. The text was indeed lacking information. Yes, the data is linearly detrended before building the regression model and then applied to non-detrended forcing data. This ensures that regression coefficients are not affected by trends that correspond to other processes. Given that we are working with the outputs from a barotropic ocean model, the trend correction, however, has only a very minor effect, as there are no other processes (such as ice-melt or steric expansion) involved. We have extended the text as follows:

“The regression model follows approaches introduced in ref. 61 and ref. 62 and estimates the response using a linear multiple stepwise regression model that fits meridional and zonal wind stress from an area of size 4 degrees (lat, lon) surrounding a tide gauge to linearly detrended tide gauge observations. The regression model is then applied to non-detrended wind stress data to capture potential trends resulting from the wind forcing.”

- Extended Data Fig. 4. Why are there only 4 principal components shown? But the paper in lines 296 - 297 proposed that the first six principal components have been used. Please clarify the rationale of the analysis.

As already outlined in the Figure caption, we have selected 4 of the 6 leading EOFs within a stepwise regression algorithm. This is why only the 4 EOFs that have finally been used are shown. The stepwise regression selects only those predictors that significantly add to the explained variance within a regression framework.

REVIEWERS' COMMENTS

Reviewer #1 (Remarks to the Author):

Review of "Acceleration of U.S. Southeast and Gulf Coast Sea-Level Rise Amplified by Internal Climate Variability" by Dangendorf et al.

The study investigates the sea level acceleration observed by tide gauge stations along the US southeast and Gulf coasts. This acceleration is unprecedented over the past 120 years and the authors show that it is induced by ocean dynamic which exceeds the forced response from historical climate model simulations. A large fraction of the residual might be attributed to wind drive Rossby waves in the tropical North Atlantic ocean indicating that the acceleration is a combination of external forcing and internal climate variability.

As I said in my previous review, I find the topic very interesting. The authors have answered all the questions I raised in the previous round. Thus, the paper has significantly improved. I can now recommend the paper for possible publication in Nature Communication.